# HALTON SCHEDULER
# FOR MASKED GENERATIVE IMAGE TRANSFORMER

**Victor Besnier**[1]    **Mickael Chen**[2,⋆]    **David Hurych**[1]

**Eduardo Valle**[2]    **Matthieu Cord**[2,3]

[1]**Valeo.ai, Prague**    [2]**Valeo.ai, Paris**    [3]**Sorbonne Université, Paris**    ⋆now at **H company, Paris**

`{firstname}.{lastname}@valeo.com`

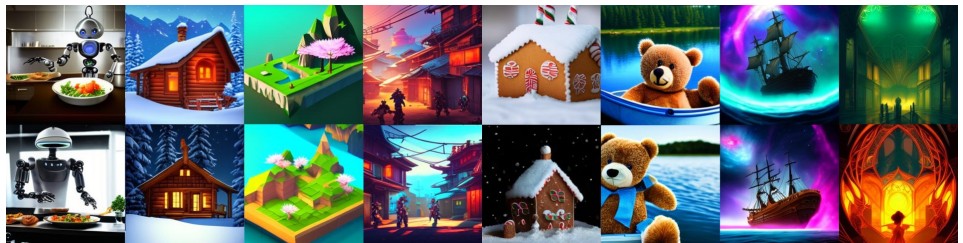

(a) **MaskGIT using our Halton scheduler.**

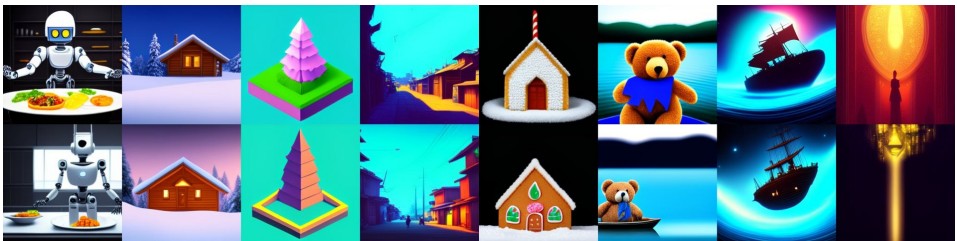

(b) **MaskGIT using the Confidence scheduler.**

Figure 1: **Text-to-Image samples comparison.** The Halton scheduler allows sampling more diverse and more detailed images than the traditional confidence scheduler, which is visible both in the foreground elements and the background.

## ABSTRACT

Masked Generative Image Transformers (MaskGIT) have emerged as a scalable and efficient image generation framework, able to deliver high-quality visuals with low inference costs. However, MaskGIT's token unmasking scheduler, an essential component of the framework, has not received the attention it deserves. We analyze the sampling objective in MaskGIT, based on the mutual information between tokens, and elucidate its shortcomings. We then propose a new sampling strategy based on our Halton scheduler instead of the original Confidence scheduler. More precisely, our method selects the token's position according to a quasi-random, low-discrepancy Halton sequence. Intuitively, that method spreads the tokens spatially, progressively covering the image uniformly at each step. Our analysis shows that it allows reducing non-recoverable sampling errors, leading to simpler hyper-parameters tuning and better quality images. Our scheduler does not require retraining or noise injection and may serve as a simple drop-in replacement for the original sampling strategy. Evaluation of both class-to-image synthesis on ImageNet and text-to-image generation on the COCO dataset demonstrates that the Halton scheduler outperforms the Confidence scheduler quantitatively by reducing the FID and qualitatively by generating more diverse and more detailed images. Our code is at `https://github.com/valeoai/Halton-MaskGIT`.

# 1 INTRODUCTION

We propose a new scheduler for Masked Generative Image Transformers (MaskGIT), an emerging alternative for image generation that offers fast inference. In contrast to the Confidence scheduler traditionally employed in MaskGIT, our Halton scheduler spreads the tokens spatially, minimizing the correlation of tokens sampled simultaneously, maximizing the information gained at each step, and ultimately resulting in more diverse and more detailed images.

Iterative methods for image synthesis, such as reverse diffusion (Betker et al., 2023; Peebles & Xie, 2023; Esser et al., 2024) or next-token prediction (Yu et al., 2022), brought significant advances in image generation, where images have attained photo-realistic quality. However, those methods impose a slow and costly inference process, and state-of-the-art methods require customizing the scheduling for the generative process (Ho et al., 2020; Song et al., 2020; Liu et al., 2021).

MaskGIT (Chang et al., 2022; 2023), a recent approach based on the Masked Auto-Encoder (He et al., 2022) paradigm, offers much faster inference times and is readily integrated into multi-modal, multitask, or self-supervised pipelines (Mizrahi et al., 2024). MaskGIT operates in a tokenized space, representing an image as a grid of discrete values. The MaskGIT training process necessitates masking a specified number of token values, letting the neural network predict their values. That framework constitutes a classification task on the discrete values of the masked tokens.

For inference, MaskGIT begins with a fully masked grid and iteratively samples the tokens according to a schedule. The scheduler aims to unmask as many tokens as possible per step to speed up the overall process while preventing sampling errors that may arise when too many tokens are unmasked in parallel.

We will show that the Confidence scheduler designed for MaskGIT (Chang et al., 2022; 2023), which unmasks the most certain tokens first, impacts the generated images' diversity and quality. That scheduler tends to select clustered tokens due to the high confidence of the area surrounding already predicted tokens (Figure 3). That results in a low information gain per sampling step.

Grounded by a mutual information analysis, we introduce a novel deterministic next-tokens scheduler that harnesses the Halton low-discrepancy sequence (Halton, 1964). Our scheduler builds upon insights on the evolution of the entropy throughout the image generation process, aiming to spread out the tokens to achieve uniform image coverage. No retraining nor noise injection is needed, and the scheduler easily plugs in MaskGIT models without requiring further changes. Compared to the traditional Confidence scheduler, we generate more diverse and higher-quality images, as visually appreciable in Figure 1 and extensively evaluated in our experiments.

Our primary contribution is the Halton scheduler (section 3), a novel scheduler for MaskGIT, with improved results without additional compute, data, or training constraints. Additionally, we present a novel mutual information analysis that clarifies the scheduler's role in MaskGIT (subsection 3.1). That analysis motivates our choice of the Halton scheduler but is not limited to our specific design and can be applied more broadly. We extensively evaluate our scheduler in the experiments presented in section 4, encompassing both class-to-image and text-to-image generation tasks.

# 2 RELATED WORK

Recently introduced, Masked Generative Image Transformers (MaskGIT) (Chang et al., 2022; 2023) follow a long line of generative models. Generative Adversarial Networks (GANs) (Radford et al., 2015; Brock et al., 2018; Karras et al., 2020; Kang et al., 2023) employed competing neural networks to improve each other. More recently, diffusion models (Dhariwal & Nichol, 2021; Song et al., 2020; Rombach et al., 2022) advanced the state of the art by reframing the problem as a simple-to-train denoising process. Concurrently, inspired by language modeling, auto-regressive image generation (Yu et al., 2022; Ramesh et al., 2021; Ding et al., 2021; Gafni et al., 2022) represents another line of attack. While both diffusion and auto-regression have demonstrated the capacity to achieve high-quality outputs, they are also prone to long inference times.

MaskGIT (Chang et al., 2022) significantly reduces inference times through the use of a bidirectional transformer encoder and a masking strategy *à la* BERT (Devlin et al., 2019). Based on the masked auto-encoder (He et al., 2022; Bao et al., 2021; Zhang et al., 2021) framework, MaskGIT directly

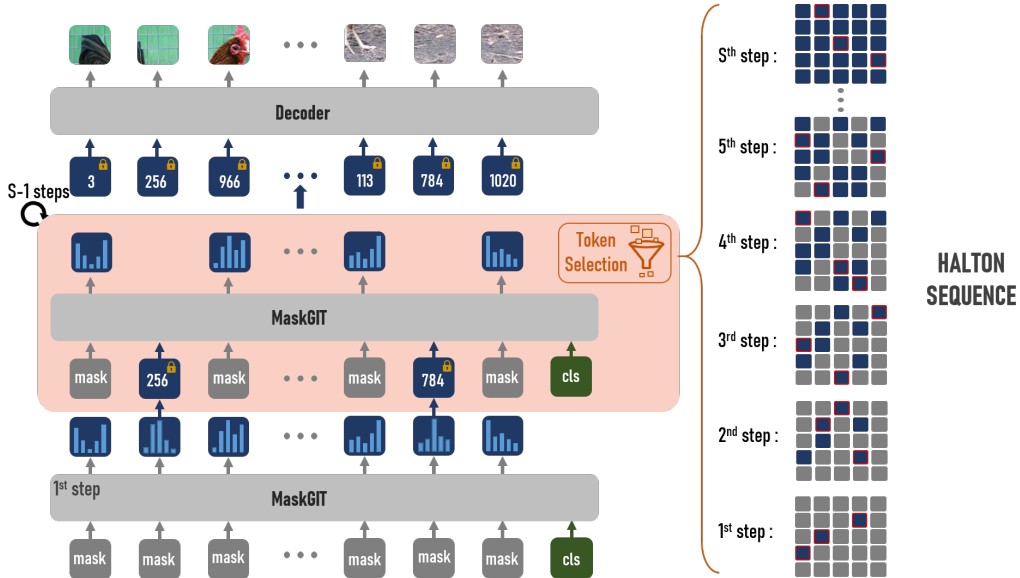

Figure 2: **(Left) MaskGIT image generation.** From masked tokens and the class token, MaskGIT samples the full image step-by-step, using a scheduler to pick which tokens to unmask. After $S$ steps, all tokens are sampled, and a deterministic decoder transforms the entire sequence into an image. **(Right) The Halton scheduler** employs the quasi-random Halton sequence to strategically distribute tokens across the image, reducing the correlation between tokens sampled in the same step and maximizing the information they provide.

inherits their scalability, native multi-modality (Mizrahi et al., 2024), and ease of integration into large-scale self-supervised pipelines (He et al., 2022; Li et al., 2023).

Modern generative models, including MaskGIT, rely on image tokenizers to reduce the image resolution, training directly in the latent space. MaskGIT, in particular, requires a discrete visual tokenizer (Van Den Oord et al., 2017; Razavi et al., 2019; Esser et al., 2020), as it is trained with a cross-entropy loss. Although discrete image tokenizers have not been as extensively studied as their continuous counterparts, recent literature indicates a growing interest in them due to their enhanced compression and superior capacity to incorporate semantic information (Yu et al., 2024; Mentzer et al., 2023b; Sun et al., 2024b).

The original MaskGIT was extended to text-to-image by MUSE (Chang et al., 2023; Patil et al., 2024), to text-to-video by MAGVIT (Yu et al., 2023a;b) and Phenaki (Villegas et al., 2023), to LiDAR point-cloud generation (Zhang et al., 2023), and even to neural simulation of interactive environments by GENIE (Bruce et al., 2024). Additionally, MaskGIT was incorporated into M2T (Mentzer et al., 2023a) with the introduction of a deterministic scheduler, 'QLDS', which employs a pseudo-random sequence. However, while M2T is designed for image compression, our work focuses on enhancing the generative capabilities of MaskGIT.

Compared to other generative models, MaskGIT and its derivatives are much less understood. Up to this point, the only demonstrated improvements came from enhancing the tokenizer (Mentzer et al., 2023b) or training a critic (Lezama et al., 2022) post-hoc, requiring additional parameters and retraining. The most recent advancements (Ni et al., 2024) target the FID as a metric by directly optimizing training and sampling hyper-parameters. This strategy is well-known for decreasing the FID without necessarily improving image quality (Barratt & Sharma, 2018; Lee & Seok, 2022; Mathiasen & Hvilshøj, 2020).

A low-discrepancy sequence, such as Halton (Halton, 1964), Sobol (Sobol, 1967), or Faure (Faure, 1981), is a deterministic sequence designed to uniformly cover a space with minimal clustering or gaps, optimizing uniformity compared to random sampling. Among all the low-discrepancy sequence improvements, the Scrambled Halton Sequence (Kocis & Whiten, 1997) improves the generated sequence in scenarios where the vanilla Halton sequence might struggle with dimension-

specific uniformity by adding a controlled amount of noise. The Leapfrog method distributes the Halton sequence across different threads, maintaining low discrepancy while allowing parallelization. However, for its simplicity and efficiency, we only investigate the Halton sequence in this work.

In this paper, we focus on the fundamental principles of the sampling process, grounded on the information gained at each step and the mutual information between tokens, while abstaining from any retraining or imposing any new assumption on the tokenizer.

## 3 METHOD

For reference, MaskGIT's sampling appears in Figure 2. In that scheme, the scheduler determines which tokens are unmasked at each step.

We seek a scheduler to generate an image with high quality and diversity as fast as possible. Formally, a schedule is an ordered list $\mathcal{S} = [\mathcal{X}_1, \mathcal{X}_2, ..., \mathcal{X}_S]$ of token sets $\mathcal{X}_s$, where $S$ is the total number of steps in the schedule. $\mathcal{X}_s$ represents the set of tokens being unmasked at step $s$, and thus $\{\mathcal{X}_s\}_{s \in 1..S}$ must be a (non-overlapping, complete) partition of the full set of tokens $\mathcal{X}$. For convenience, we denote as $\mathcal{X}_{<s}$ the set of tokens already unmasked at the start of step $s$, as $\mathcal{X}_{>s}$ the set of tokens yet to be unmasked at the end of step $s$, and as $n_s = |\mathcal{X}_s|$ the number of tokens in $\mathcal{X}_s$. We also denote as $X^i$ the $i$-th token in $\mathcal{X}$ and as $X_s^i$ the $i$-th token in subset $\mathcal{X}_s$, arbitrarily ordered.

Next, we thoroughly analyze MaskGIT's sampling (Chang et al., 2022) under the lens of Mutual Information (MI) of sampled tokens. Based on that analysis, we present a new scheduler built on the Halton sequence (Halton, 1964).

### 3.1 THE ROLE OF MUTUAL INFORMATION

The goal of the generative model is to sample from $p(\mathcal{X})$, the joint distribution of the tokens that constitute the data. For MaskGIT (Chang et al., 2022), this distribution decomposes according to the schedule $\mathcal{S}$:

$$p(\mathcal{X}) = \prod_{s=1}^{S} p(\mathcal{X}_s | \mathcal{X}_{<s}) = p(\mathcal{X}_1) p(\mathcal{X}_2 | \mathcal{X}_1) p(\mathcal{X}_3 | \mathcal{X}_2, \mathcal{X}_1) \dots p(\mathcal{X}_S | \mathcal{X}_{<S}) \tag{1}$$

Thus, at any individual step $s$, we aim to sample from the joint distribution of all considered tokens:

$$p(\mathcal{X}_s | \mathcal{X}_{<s}) = p(X_s^1, X_s^2, \dots, X_s^{n_s} | \mathcal{X}_{<s}). \tag{2}$$

However, MaskGIT is trained on the cross-entropy loss for each token *individually*, and, therefore, only provides an estimate of each token marginal distribution $p(X_s^i | X_{<s})$. Therefore, MaskGIT only samples from the product distribution:

$$\prod_{i=1}^{n_s} p(X_s^i | \mathcal{X}_{<s}) = p(X_s^1 | \mathcal{X}_{<s}) p(X_s^2 | \mathcal{X}_{<s}), \dots, p(X_s^{n_s} | \mathcal{X}_{<s}). \tag{3}$$

In summary, MaskGIT models the product distribution, which is a loose stand-in for the joint distribution we need. The sampling process must, therefore, reduce the gap between those two distributions, measured by their Kullback-Leibler divergence, which can be interpreted as the Mutual Information of all $X_s^i$ after knowing $\mathcal{X}_{<s}$:

$$\text{MI}(\mathcal{X}_s | \mathcal{X}_{<s}) = \text{D}_{\text{KL}} \left( p(\mathcal{X}_s | \mathcal{X}_{<s}) \,\middle\|\, \prod_{i=1}^{n_s} p(X_s^i | \mathcal{X}_{<s}) \right) \tag{4}$$

where $\mathcal{X}_s | \mathcal{X}_{<s}$ is a shorthand for $X_s^1, X_s^2, \dots, X_s^{n_s} | \mathcal{X}_{<s}$.

Our goal is to propose a schedule that minimizes the Mutual Information aggregated over the inference steps $\sum_{s=1}^{S} \text{MI}(\mathcal{X} | \mathcal{X}_{<s})$, which we decompose as a sum of entropies conditioned on the unmasked tokens:

$$\sum_{s=1}^{S} \text{MI}(\mathcal{X}_s | \mathcal{X}_{<s}) = \sum_{s=1}^{S} \underbrace{\sum_{i=1}^{n_s} \text{H}(X_s^i | \mathcal{X}_{<s})}_{(a)} + \sum_{s=1}^{S} \underbrace{\sum_{i=1}^{n_s} -\text{H}(\mathcal{X}_s | \mathcal{X}_{<s}, X_s^i)}_{(b)}. \tag{5}$$

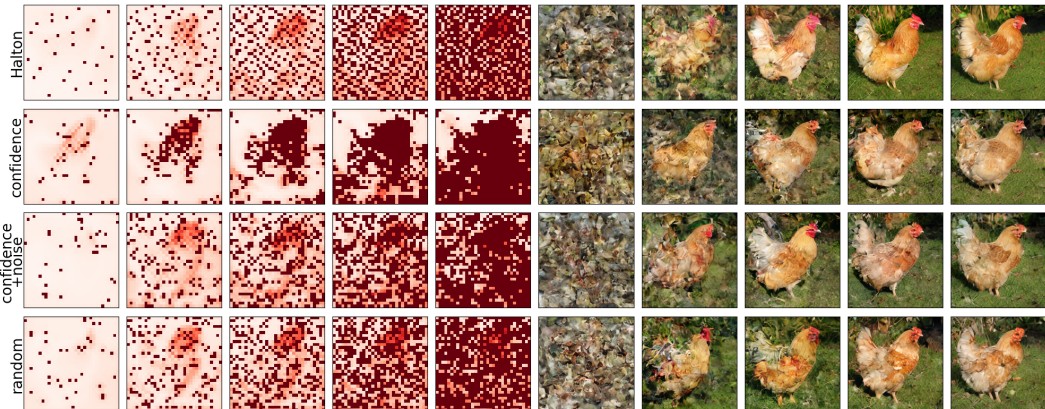

Figure 3: **Evolving predictions of different schedulers on a reconstruction comparison.** (left) MaskGIT predicted entropy maps, with darker colors for lower entropies and known tokens in dark red. Images are reconstructed (right) by revealing the ground-truth tokens of a reference image at the locations selected by the scheduler. The Confidence scheduler picks the most certain tokens first and, thus, tends to cluster around already unmasked areas. Adding noise alleviates but does not solve the problem. The Halton scheduler provides a more uniform image coverage at each step of the sampling process.

The term Eq. 5.a corresponds to the entropy of the tokens to be predicted, whose uncertainty in $\mathcal{X}_s$ we strive to minimize. We can interpret the term Eq. 5.b as the amount of reciprocal information the tokens of $\mathcal{X}_s$ share, which we should also minimize. Intuitively: because we sample the $X_s^i$ tokens from their marginal distribution (as explained above), we assume them to be independent. When that assumption is broken, and tokens in the same step have high mutual information, we risk creating incompatibilities. For instance, there is a risk of two petals of the same flower having conflicting shapes. Note that, in theory, ramping up the scheduling inference steps $n_s$ should reduce the overall error (Equation 5), which should, in turn, improve the generation quality. We will test that hypothesis in the next section.

While an estimate of Eq. 5.a is readily available from MaskGIT's output, Eq. 5.b is computationally intractable. Instead, we exploit the strong correlation, for natural images, between pixel color similarity and spatial distance, which we extrapolate as the assumption that the conditional entropy $H(X_i|X_j)$ between pairs of visual tokens $X_i$ and $X_j$, with spatial coordinates $\boldsymbol{x}_i$ and $\boldsymbol{x}_j$, should be strongly correlated to their Euclidean distance $||\boldsymbol{x}_i - \boldsymbol{x}_j||_2$. While this assumption is well-established in the pixel space (Huang & Mumford, 1999), we demonstrate in the Appendix that this principle also extends the compressed space of the tokenizer. This leads us to low discrepancy sequences for scheduling tokens sampling, which we describe next.

## 3.2 LOW DISCREPANCY HALTON SEQUENCE

The Halton sequence (Halton, 1964) is a low-discrepancy sequence, *i.e.*, a sequence of points filling a space, with the property that the number of points falling into a subset of the space is proportional to the measure of that subset. For us, the relevant properties are that those sequences are quasi-random and that consecutive points on the sequence tend to be distant in space. Those properties will help us to sample tokens as independently as possible, keeping the term Eq. 5.b small. Moreover, because the sequence ensures uniform spatial coverage of the image, for any step $s$, the known tokens $\mathcal{X}_{<s}$ give us a reasonable estimate of the whole image, also limiting the growth of term Eq. 5.a.

Formally, let the Halton sequence $\mathcal{H}^n = [x_1, x_2, \cdots, x_n]$ be a sequence of $n$ points $x_i \in \mathbb{R}^d$, and let $i \in \{1, \cdots, n\}$ be the index of the point to sample written in the $Radix\ b$ notation:

$$i = a_m b^m + a_{m-1} b^{m-1} + \cdots + a_2 b^2 + a_1 b + a_0, \tag{6}$$

where $b > 1$ is called the base, $0 < a_l < b$ and $l \in \{0, 1, \ldots, m-1\}$. The Halton sequence uses the R-inverse function of $i$:

$$\Phi_b(i) = a_0 b^{-1} + a_1 b^{-2} + \cdots + a_{m-1} b^{-m-2} + a_m b^{-m-1}. \tag{7}$$

We sample the next token's position from a 2D Halton sequence with bases 2 and 3, storing them as an ordered list of pairs

$$\mathcal{H}^{n_h} = [(\Phi_2(1), \Phi_3(1)), (\Phi_2(2), \Phi_3(2)), \ldots, (\Phi_2(n), \Phi_3(n))], \tag{8}$$

where $n_h$ is strictly greater than the total number of tokens $n$ to predict.

In practice, we discretize the sequence $\mathcal{H}^{n_h}$ into a grid corresponding to the token map. We discard duplicate grid coordinates and ensure all coordinates are sampled. Finally, we use the $\mathcal{H}^n$ list, where each index corresponds to one distinct token. A detailed algorithm for the Halton sequence appears in the Appendix.

### 3.3 Properties of Sampling Methods

We have identified in subsection 3.1 the fundamental quantities necessary to minimize the discrepancy between the joint probability we aim to sample from and the approximation available with MaskGIT (Equation 4). Then, subsection 3.2 introduced our Halton scheduler as a sampling method. We discuss here how it behaves in terms of conditional entropies of Equation 5, in particular, compared to the Confidence scheduler from the original MaskGIT (Chang et al., 2022).

First, we note that the Confidence scheduler is focused on minimizing term Eq. 5.a greedily at each step: by only selecting the top-k tokens in terms of confidence, they effectively minimize the entropy of those tokens. As presented in the original paper, confidence sampling does not account for longer-term gains. In fact, we show in Figure 3 that each step does not bring as much new information as other schedulers, leaving some high entropy regions for the few last steps. It also neglects the mutual information of the selected tokens. Such behavior is visible in Figure 4, which tracks metrics for $\mathcal{X}_s$ at each step. Their entropy stays low but increases sharply at the end, while the distance to each other or to seen tokens, which should be high, as explained in subsection 3.1, remains very low.

As a result, confidence sampling performs poorly in our experiments. Examination of the inference code reveals that the sampling method introduces Gumbel Noise in the confidence score[1], on top of the top-k selection mechanism and the temperature of the softmax, when selecting the tokens to unmask (Besnier & Chen, 2023), an enhancement not discussed in the original publication. The noise is used only to select which tokens to unmask and does not affect the token value itself. Adding noise effectively reduces the greediness of the algorithm and the over-focus on the term Eq. 5.a, allowing for a better balance across time steps and reducing term Eq. 5.b. However, looking at Figure 3 and Figure 4, we still see a spike of entropy for the latest steps by the end of the generation, which reveals that an ideal tuning of the noise scheduler is challenging to obtain. That results in an undesirable saturation, and even deterioration, of the performance of the noised confidence scheduler as the number of steps grows (see Figure 5).

Our Halton Scheduler uses fixed order for the tokens to make sure that subsets $\mathcal{X}_{<s}$, $\mathcal{X}_s$, and their union are all well distributed across the image, thus controlling all the terms of the mutual information in Equation 5, as explained in the previous subsection. That is visible as the slow progression of the metrics in Figure 4, linearly distributed across generation steps. That indicates a better compromise between both quantities (a) and (b) from Equation 4. Our scheduler handles entropy gain at each step better. Moreover, with the Halton scheduler, the performance of MaskGIT keeps improving with increasing inference steps, as shown in Figure 5. Intuitively, as seen in Figure 3, the overall composition of the image is then decided early in the sampling process, while later stages add high-frequency information.

Our analysis assumes that the training results in a model that accurately estimates the marginal distributions for each token. Although such an assumption is common, it does not account for potential miscalibration or other biases. While those issues could impact our analysis, we find our predictions consistent with our experimental results, as demonstrated in section 4.

## 4 Experiments

This section presents a comprehensive evaluation of our method, focusing on the enhancements brought by our Halton scheduler in image quality and diversity compared to the baseline Confidence

---

[1]In the original code, a linear decay is employed.

| Models | Scheduler | Steps | #Para. | FID ↓ | IS ↑ | Precision↑ | Recall↑ |
|--------|-----------|-------|--------|-------|------|-----------|---------|
| Our MaskGIT | Confidence | 32 | 484M | 7.5 | 227.2 | **0.76** | 0.56 |
| | Random | 32 | 484M | 9.7 | 196.2 | 0.75 | 0.51 |
| | Halton (ours) | 32 | 484M | **5.3** | **229.2** | **0.76** | **0.63** |
| Our MaskGIT | Confidence | 12 | 705M | 6.80 | 181.2 | **0.80** | 0.49 |
| | Random | 12 | 705M | 7.90 | 156.3 | 0.76 | 0.45 |
| | Halton (ours) | 12 | 705M | **6.19** | **184.2** | 0.79 | **0.50** |
| Our MaskGIT | Confidence | 32 | 705M | 5.93 | **301.1** | **0.87** | 0.46 |
| | Random | 32 | 705M | 6.33 | 276.0 | 0.73 | 0.49 |
| | Halton (ours) | 32 | 705M | **3.74** | 279.5 | 0.81 | **0.60** |

Table 1: **Ablation study showcasing the relevance of the Halton scheduler on ImageNet 256×256..** A comparison of our own MaskGIT methods with different schedulers shows that the Halton Scheduler outperforms the others.

scheduler. We present qualitative and quantitative results on two distinct tasks, each using different modalities: class-to-image ( subsection 4.1) and text-to-image ( subsection 4.2).

## 4.1 CLASS-TO-IMAGE SYNTHESIS

For our experiments in class-conditional image generation, we used the ImageNet dataset (Deng et al., 2009), consisting of 1.2 million images across 1,000 classes. We evaluated our approach using key metrics commonly employed in class-conditional generative modeling: Fréchet Inception Distance (FID) (Heusel et al., 2017), Inception Score (IS) (Salimans et al., 2016), as well as Precision and Recall (Kynkäänniemi et al., 2019).

Our Masked Generative Image Transformer model was trained on ImageNet at a resolution of $256 \times 256$, leveraging the pre-trained VQ-GAN from (Sun et al., 2024b) with a codebook size of 16,384 and a down-scaling factor of 8. We incorporated the class condition into a ViT-XL and ViT-L (Peebles & Xie, 2023) architecture via Adaptive Layer Norm, and the input was patchified with a factor of 2. Further architectural details, hyper-parameters, and training configurations are provided in the Appendix.

We underscore the core contribution of our Halton Scheduler in Table 2 and Table 1, demonstrating its effectiveness across a variety of sce-

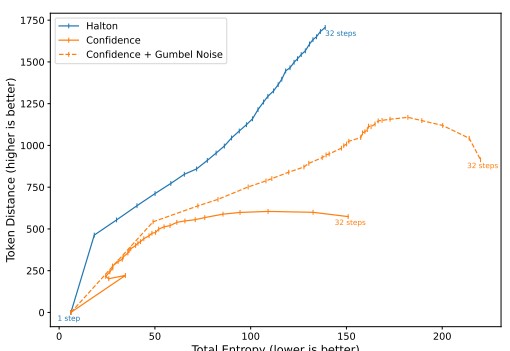

Figure 4: **Mutual Information scheduler analysis.** (x-axis) The sum of the entropy of the unmasked tokens in $\mathcal{X}_s$ corresponds to term Eq. 5.a. (y-axis) The sum of the distances of each unmasked token in $\mathcal{X}_s$ to their closest neighbor is a reversed proxy for the term Eq. 5.b. Each scheduler draws a curve in this plot as the number of sampling steps progresses. The Halton scheduler stays closer to the top-left corner, the desirable part of the plot.

narios, including different resolutions, network architectures, tokenizers, and number of steps. We benchmark the Halton Scheduler against the original Confidence Scheduler and a baseline Random Scheduler, which unmasks tokens at random locations. Results highlight the superiority of our method: with our MaskGIT model on ImageNet 256×256, the Halton Scheduler achieves a $-2.19$ FID improvement over the Confidence Scheduler from (Chang et al., 2022), validating its efficiency and robustness. On ImageNet 512×512, due to the incomplete availability of the original authors' code, we employed a publicly available reproduction (Besnier & Chen, 2023). Our Halton Scheduler achieves strong performance $-2.27$ FID using the pre-trained MaskGIT model with no retraining and 32 steps.

| Models | Scheduler | Steps | #Para. | FID ↓ | IS ↑ | Precision↑ | Recall↑ |
|---|---|---|---|---|---|---|---|
| MaskGIT (Besnier & Chen, 2023) | Confidence | 12 | 246M | 7.60 | 185.1 | **0.81** | 0.49 |
| | Random | 12 | 246M | 18.84 | 132.1 | 0.69 | 0.48 |
| | Halton (ours) | 12 | 246M | **7.15** | **186.4** | 0.80 | **0.55** |
| MaskGIT (Besnier & Chen, 2023) | Confidence | 32 | 246M | 8.38 | 180.6 | 0.79 | 0.49 |
| | Random | 32 | 246M | 10.47 | 174.3 | 0.75 | 0.52 |
| | Halton (ours) | 32 | 246M | **6.11** | **184.0** | **0.80** | **0.57** |

Table 2: **Ablation showcasing the relevance of the Halton scheduler on ImageNet 512×512..** A comparison of open-source MaskGIT with different schedulers shows that the Halton Scheduler outperforms other schedulers, particularly a purely random scheduler.

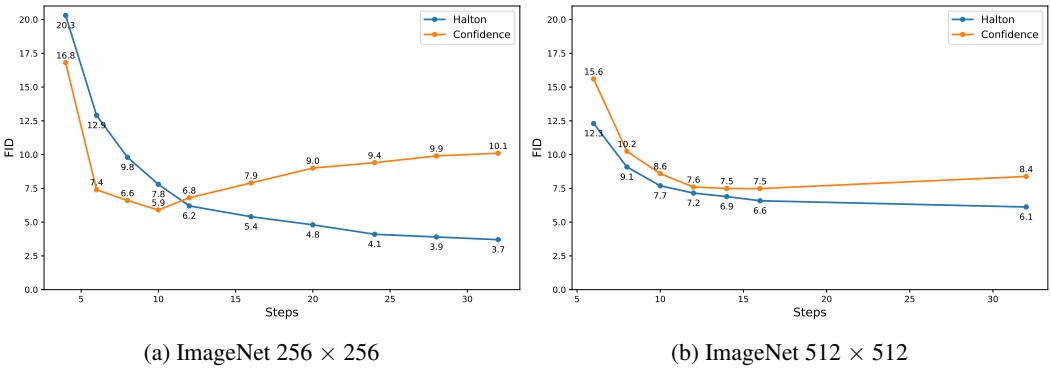

(a) ImageNet 256 × 256          (b) ImageNet 512 × 512

Figure 5: **Ablation on the number of steps.** The Halton scheduler demonstrates scalability as the number of steps increases, whereas the Confidence scheduler's performance deteriorates. The Halton scheduler consistently outperforms the Confidence scheduler when the number of steps exceeds 12.

In Figure 5, we demonstrate the impact of varying the number of steps using both our models on ImageNet 256×256 and ImageNet 512×512, using the pre-trained models from the MaskGIT (Besnier & Chen, 2023) reproduction. The Halton scheduler improves performance when the number of steps is above 12. Moreover, it exhibits better scaling properties as the number of sampling steps increases. In contrast, the Confidence scheduler decreases performance as the number of steps increases.

To provide a clearer perspective of our model within the broader landscape of generative models, we compare our implementation of MaskGIT in Table 3 against other image synthesis methods and demonstrate that our approach outperforms the original MaskGIT implementation, reducing the FID score by almost 40% and improving the IS by over 50%. In addition, our results are competitive with other SOTA Masked Image Modeling (MIM) techniques without requiring extensive optimization directly targeting the FID of AutoNAT (Ni et al., 2024).

| Type | Model | #Para. | FID↓ | IS↑ | Precision↑ | Recall↑ |
|---|---|---|---|---|---|---|
| GAN | StyleGan-XL (Sauer et al., 2022) | 166M | 2.30 | 265.1 | 0.78 | 0.53 |
| Diffusion | DiT-XL/2 (Peebles & Xie, 2023) | 675M | 2.27 | 278.2 | 0.83 | 0.57 |
| AR | LlamaGen-3B (Sun et al., 2024b) | 3.1B | 2.18 | 263.3 | 0.81 | 0.58 |
| MIM | MaskGIT (Chang et al., 2022) | 227M | 6.18 | 182.1 | 0.80 | 0.51 |
| | Token-Critics (Lezama et al., 2022) | — | 4.69 | 174.5 | 0.76 | 0.53 |
| | AutoNAT-L (Ni et al., 2024) | 194M | **2.68** | 278.8 | — | — |
| | FSQ (Mentzer et al., 2023b) | — | 4.53 | — | **0.86** | 0.45 |
| | MAGE (Li et al., 2023) | 439M | 7.04 | 123.5 | — | — |
| | **Ours + Halton** | 705M | 3.74 | **279.5** | 0.81 | **0.60** |

Table 3: **Model comparison on class-conditional ImageNet 256×256 benchmark.** The proposed Halton scheduler using our own implementation of MaskGIT outperforms the original, demonstrating competitive performance among Masked Image Modeling (MIM) approaches. Full table in Appendix.

| Type | Model | #Para. | #Train Img | FID↓ | IS↑ | Prec.↑ | Recall↑ | Clip↑ |
|------|-------|--------|-----------|------|-----|--------|---------|-------|
| Diff.‡ | LDM 2.1 (Rombach et al., 2022) | 860M | 3,900M | 9.1 | – | – | – | – |
| | Wurstchen (Pernias et al., 2024) | 990M | 1,400M | 23.6 | – | – | – | – |
| | PixArt-$\alpha$ (Chen et al., 2024) | 610M | 25M* | **7.3** | – | – | – | – |
| | MicroDiT (Sehwag et al., 2024) | 1,200M | 37M | 12.7 | – | – | – | – |
| MIM† | aMused (Patil et al., 2024) | 603M | 1,120M | 38.9 | 23.5 | 0.52 | 0.37 | **25.7** |
| | **Ours (Rand)** | 480M | 17M | 33.3 | 24.8 | 0.57 | 0.41 | 25.0 |
| | **Ours (Conf)** | 480M | 17M | 31.5 | 25.4 | 0.59 | 0.43 | 25.4 |
| | **Ours (Halton)** | 480M | 17M | **28.8** | **26.6** | **0.61** | **0.47** | **25.7** |

Table 4: **Model comparisons on text-to-image zero-shot COCO**. The Halton scheduler significantly enhances image fidelity over the Confidence sampler. It also achieves competitive results with much fewer training images than diffusion models. ‡ Open-source Diffusion computed using resolution 512×512 and FID-30k. † Open-source MIM computed using FID-10k and resolution 256×256. * uses 10M private images.

## 4.2 Text-to-Image Synthesis

For the text-to-image generation experiments, we employed a combination of real-world datasets, including CC12M (Changpinyo et al., 2021) and a subset of Segment Anything (Kirillov et al., 2023), as well as synthetic datasets such as JourneyDB (Sun et al., 2024a) and DiffusionDB (Wang et al., 2022). In total, we gathered approximately 17 million images. We evaluated our method using the key metrics for text-to-image generative models on the zero-shot COCO dataset (Lin et al., 2014): Fréchet Inception Distance (FID) (Heusel et al., 2017), CLIP-Score (Radford et al., 2021), Precision, and Recall (Kynkäänniemi et al., 2019).

Our MaskGIT model was trained exclusively on publicly available datasets at a resolution of 256. The same frozen VQGAN (Sun et al., 2024b) model was used for class conditioning, and a frozen T5-XL (Chung et al., 2024) model for text token embeddings. To incorporate text conditions, we employed cross-attention mechanisms. Further details regarding the model architecture, hyperparameters, and training specifics can be found in the Appendix.

In Table 4, we present zero-shot results on the COCO dataset compared with other open-source text-to-image synthesis methods. We benchmark against aMused (Patil et al., 2024), the only open-source MaskGIT Image Transformer with text conditioning. In this evaluation, our approach not only surpasses aMused by a notable margin, achieving a reduction of 10.1 in FID, but also demonstrates that the Halton scheduler improves model quality at no additional computational cost compared to the Confidence scheduler, with a further reduction of 2.7 in FID. Although masked generative transformers currently underperform in comparison to diffusion-based approaches, remark that those methods employ larger datasets with more trainable parameters.

Examples generated by our method are illustrated in Figure 7 and Figure 6, showcasing curated samples. Despite using only publicly available data and a relatively limited number of parameters, our models can generate a diverse range of high-quality samples. As demonstrated in Figure 1, a qualitative comparison between the two schedulers reveals that our model employing the Halton scheduler exhibits a notable ability to generate images with intricate details and a high degree of diversity. Compared to the Confidence scheduler, the Halton scheduler is particularly adept at producing images with enhanced sharpness and a more diverse range of backgrounds. Further evidence on the class-to-image model can be found in the Appendix.

## 5 Conclusion

In this paper, we introduced the Halton scheduler for the sampling of Masked Generative Image Transformers. Grounded on the mutual information of the sampled tokens at each step, we demonstrated that the original Confidence scheduler performs inadequately, requiring the addition of Gumbel Noise during the sampling process to mitigate its spatially clustering tendencies, and even so, only partially succeeding. In contrast, based on the low-discrepancy Halton sequence, the Halton scheduler effectively spreads the selected tokens spatially, eliminating the need for additional noise injection or training. The Halton scheduler improves the quality and diversity of both class-to-image and text-to-image generation while keeping MaskGIT's inference speed high.

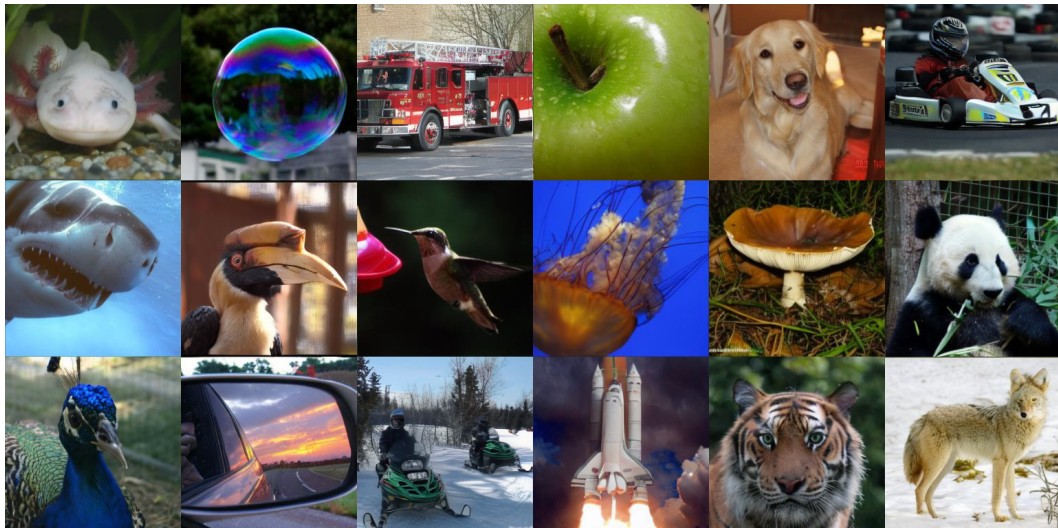

Figure 6: **Class-to-Image curated example on Imagenet $256 \times 256$.** Images generated using the Halton scheduler show good visual quality across classes, demonstrating its effectiveness. Additional samples, randomly selected, are available in the Appendix.

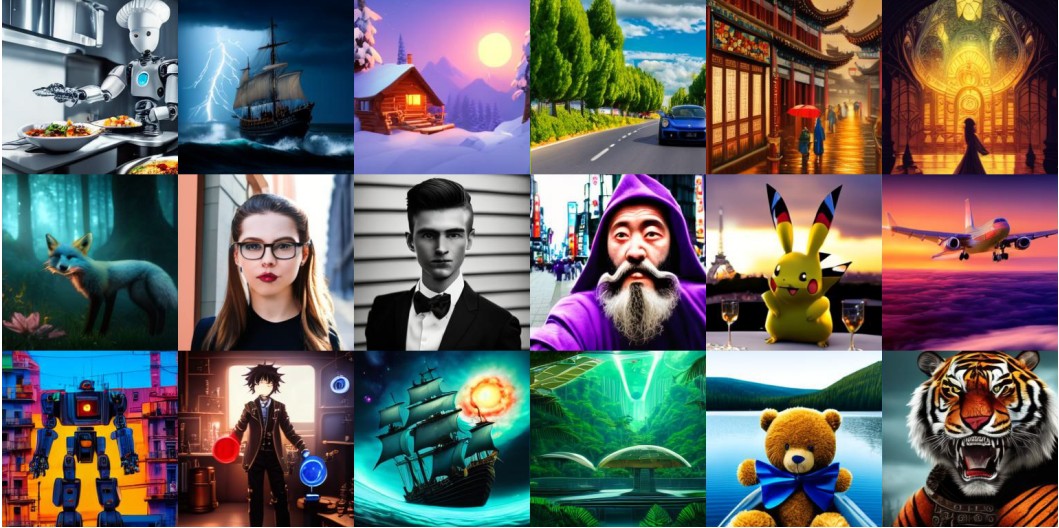

Figure 7: **Text-to-Image curated examples.** Images generated using the Halton scheduler show good visual quality across diverse textual prompts. Prompts are available in the Appendix.

The Halton scheduler enhances the performance of MaskGIT by reducing the correlation between tokens by breaking short-range dependencies. One current limitation of the scheme and of MaskGIT is the inability to accommodate long-range dependencies. Solving that issue while maintaining the advantageous inference times of MaskGIT represents an exciting research frontier. It may require rethinking the image's encoding, scheduling, and decoding as a whole integrated framework.

**Acknowledgments** This research received the support of EXA4MIND project, funded by the European Union's Horizon Europe Research and Innovation Programme under Grant Agreement N°101092944. Views and opinions expressed are, however, those of the authors only and do not necessarily reflect those of the European Union or the European Commission. Neither the European Union nor the granting authority can be held responsible for them. We acknowledge EuroHPC Joint Undertaking for awarding us access to Karolina at IT4Innovations, Czech Republic.

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

# 6 TRAINING DETAILS

Table 5 provides all the hyperparameters used to train our models across both modalities. In Table 6, we detail the architecture of our models.

For class-to-image generation, we employed an architecture similar to DiT-XL (Peebles & Xie, 2023), utilizing a patch size of 2 to reduce the number of tokens from 32×32 to 16×16. Due to GPU memory constraints, we opted not to use Exponential Moving Averages (EMA). Additionally, we used the 'tie_word_embedding' technique, where the input and output layers share weights, reducing the number of trainable parameters.

We used the T5-XL encoder for text-to-image synthesis, which processes 120 text tokens per input, resulting in a text embedding of size [120, 2048] for each sentence. To integrate text conditioning, we employed a transformer architecture similar to DiT-L (Peebles & Xie, 2023), the largest model we could fit on our GPU with EMA. The condition is incorporated using classical cross-attention.

| Condition | text-to-image | class-to-image |
|---|---|---|
| Training steps | $5 \times 10^5$ | $2 \times 10^6$ |
| Batch size | 2048 | 256 |
| Learning rate | $5 \times 10^{-5}$ | $1 \times 10^{-4}$ |
| Weight decay | 0.05 | $5 \times 10^{-5}$ |
| Optimizer | AdamW | AdamW |
| Momentum | $\beta_1 = 0.9, \beta_2 = 0.999$ | $\beta_1 = 0.9, \beta_2 = 0.96$ |
| Lr scheduler | Cosine | Cosine |
| Warmup steps | 2500 | 2500 |
| Gradient clip norm | 0.25 | 1 |
| EMA | 0.999 | − |
| CFG dropout | 0.1 | 0.1 |
| Data aug. | No | Horizontal Flip |
| Precision | bf16 | bf16 |

Table 5: Hyper-parameters used in the training of text-to-img and class-to-img models.

| Condition | text-to-image | class-to-image |
|---|---|---|
| Parameters | 479.8M | 705.0M |
| Input size | $32 \times 32$ | $32 \times 32$ |
| Hidden dim | 1024 | 1152 |
| Codebook size | 16384 | 16384 |
| Depth | 24 | 28 |
| Heads | 16 | 16 |
| Mlp dim | 4096 | 4608 |
| Patchify (p=) | 2 | 2 |
| Dropout | 0.0 | 0.0 |
| Conditioning | Cross-attention | AdaLN |

Table 6: Architecture design of the text-to-img and class-to-img models.

# 7 EUCLIDEAN DISTANCE IN THE TOKEN SPACE

One assumption of our analysis is that parts of the image that are closer together tend to be more similar in appearance. This relationship is well understood in pixel space (Huang & Mumford, 1999). In Figure 8, we show that the principle also holds in the token space. We measure the appearance dissimilarity of tokens using the Euclidean distance of their corresponding latent representation on the LlamaGen tokenizer on the ImageNet dataset. As shown in the figure, tokens that are spatially

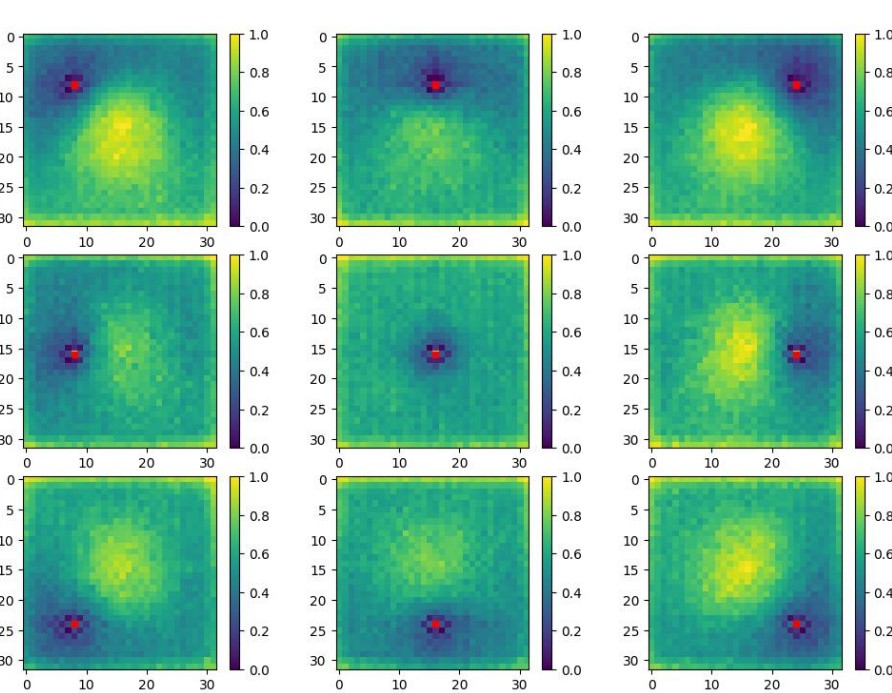

Figure 8: **Spatial distance vs. appearance dissimilarity of tokens.** The color map indicates the normalized appearance dissimilarity between each token and the reference token (shown in red). The closest tokens to the reference token are the most similar (dark blue). Tokens further away tend to be dissimilar (bright yellow).

close to the reference token (in red) also have the closest representations in feature space (dark blue). Tokens that are spatially further apart tend to have dissimilar representations (green to yellow).

## 8 GENERATIVE METHOD COMPARISON

We show here a complete evaluation of our methods with the Halton scheduler again recent methods from the literature in Table 7. The analysis of Figure 9 shows that we are narrowing the gap between diffusion, auto-regressive, and masked image modeling, bringing the latter to the competitive forefront of generative methods while maintaining their speed advantage (dozens of inference steps for MIM *vs.* hundreds for auto-regressive and for diffusion).

## 9 INTERMEDIATE GENERATION

An interesting property of our approach is shown in Figure 10 depicting the intermediate construction of the macaw (**088**). It highlights that most images are already fixed after a few steps. First, the bird's blue color and the white background are completely set after only four steps with $25/1024 \approx 2\%$ unmasked tokens. The shape is fixed at the 8th step (8% tokens unmasked), and the texture starts to appear at 12 steps (16% tokens unmasked). This means that the rest of the token will only influence the high-frequency details of the generated image. We push the analysis further by computing the FID and IS for these intermediate samples (see Table 8), where we evaluate the results given the generated intermediate images. While the first 16 steps significantly increase both the FID and the IS, the last 12 steps only decrease the FID score by 0.14 points.

| Type | Model | #Para. | FID↓ | IS↑ | Precision↑ | Recall↑ |
|---|---|---|---|---|---|---|
| GAN | BigGAN (Brock et al., 2018) | 112M | 6.95 | 224.5 | **0.89** | 0.38 |
| | GigaGAN (Kang et al., 2023) | 569M | 3.45 | 225.5 | 0.84 | **0.61** |
| | StyleGan-XL (Sauer et al., 2022) | 166M | **2.30** | **265.1** | 0.78 | 0.53 |
| Diffusion | ADM (Dhariwal & Nichol, 2021) | 554M | 10.94 | 101.0 | 0.69 | **0.63** |
| | CDM (Ho et al., 2020) | — | 4.88 | 158.7 | — | — |
| | LDM-4 (Rombach et al., 2022) | 400M | 3.60 | 247.7 | — | — |
| | DiT-XL/2 (Peebles & Xie, 2023) | 675M | **2.27** | **278.2** | **0.83** | 0.57 |
| AR | VQGAN (Esser et al., 2020) | 1.4B | 15.78 | 74.3 | — | — |
| | ViT-VQGAN (Yu et al., 2021) | 1.7B | 4.17 | 175.1 | — | — |
| | RQTran.-re(Lee et al., 2022) | 3.8B | 3.80 | **323.7** | — | — |
| | LlamaGen-3B (Sun et al., 2024b) | 3.1B | **2.18** | 263.3 | **0.81** | **0.58** |
| MIM | MaskGIT (Chang et al., 2022) | 227M | 6.18 | 182.1 | 0.80 | 0.51 |
| | Token-Critics (Lezama et al., 2022) | — | 4.69 | 174.5 | 0.76 | 0.53 |
| | AutoNAT-L (Ni et al., 2024) | 194M | **2.68** | 278.8 | — | — |
| | FSQ (Mentzer et al., 2023b) | — | 4.53 | — | **0.86** | 0.45 |
| | MAGE (Li et al., 2023) | 439M | 7.04 | 123.5 | — | — |
| | **Ours + Halton** | 705M | 3.74 | **279.5** | 0.81 | **0.60** |

Table 7: **Model comparison on class-conditional ImageNet 256×256 benchmark.** The proposed Halton scheduler outperforms the original MaskGIT considerably, demonstrating competitive performance among Masked Image Modeling (MIM) approaches.

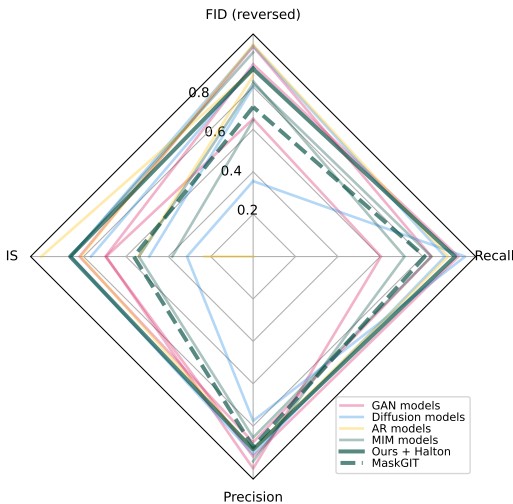

Figure 9: **Analysis of the SOTA results of Table 7.** In this radar plot, each model is represented as a line. Each metric is normalized to 1 for its best model. The FID is reversed, so higher is always better. The improvement brought by the Halton scheduler over the vanilla MaskGIT with Confidence scheduler is immediately noticeable. Our scheduler brings fast masked generative transformers to the competitive vanguard of existing methods.

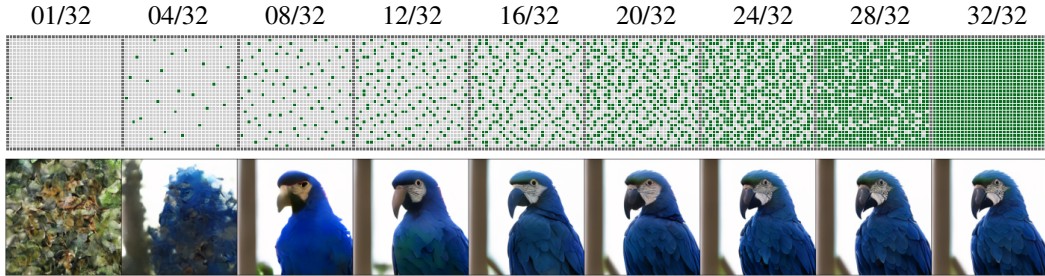

Figure 10: **Evolution of the sampling using Halton scheduler.** The macaw's (`088`) color, texture, and shape, as well as the background, are set after only 12 steps, with only ∼16% tokens predicted. That showcases the ability of the Halton scheduler to extract information from the tokens by reducing their correlation.

| Steps | Percentage of tokens | FID ↓ | IS ↑ |
|-------|----------------------|-------|------|
| 4/32  | 2%   | -     | -     |
| 8/32  | 8%   | 146.2 | 7.31  |
| 12/32 | 16%  | 76.7  | 24.1  |
| 16/32 | 28%  | 24.9  | 79.6  |
| 20/32 | 43%  | 8.85  | 142.2 |
| 24/32 | 61%  | 6.25  | 174.3 |
| 28/32 | 82%  | 6.12  | 182.2 |
| 32/32 | 100% | 6.11  | 184.0 |

Table 8: **Evaluation of intermediate generated samples on ImageNet 512×512.** Most of the gains are on early steps, which are crucial to achieving good FID and IS. Later steps keep improving but may be skipped as a compromise between quality and compute.

## 10 PSEUDO-CODE FOR HALTON SEQUENCE

In algorithm 1, we detail the generation of the Halton sequence, producing a sequence of size $n'$ with a base $b$. In practice, we generate two sequences with $b = 2$ and $b = 3$, respectively, representing 2D coordinates of the points to select. We then discretize the space in a $32 \times 32$ grid. Duplicate points are discarded, ensuring complete grid coverage by setting $n'$ appropriately. The coordinates of the remaining points determine the order of token unmasking during sampling.

## 11 TEXT PROMPTS

Prompts used for our text-to-image model, corresponding to Figure 7 in the main paper, from top-left to bottom-right:

1. A robot chef expertly crafts a gourmet meal in a high-tech futuristic kitchen, intricate details.

2. An old-world galleon navigating through turbulent ocean waves under a stormy sky lit by flashes of lightning.

3. A cozy wooden cabin perched on a snowy mountain peak, glowing warmly in the night, styled like a classic Disney movie, featured on ArtStation.

4. A blue sports car is parked. The sky above is partly cloudy, suggesting a pleasant day. The trees have a mix of green and brown foliage. There are no people visible in the image.

5. An oil painting of rain in a traditional Chinese town.

6. Volumetric lighting, spectacular ambient lights, light pollution, cinematic atmosphere, Art Nouveau style illustration art, artwork by SenseiJaye, intricate detail.

7. A mystical fox in an enchanted forest, glowing flora, and soft mist, rendered in Unreal Engine.

---

**Algorithm 1:** Compute the Halton sequence

1 **Parameters:**
2     $b$: the base of the Halton sequence,
3     $n'$: the number of points in the sequence to compute
4 **Results:**
5     $S$: the first $n'$ points of the Halton sequence in base $b$

6 $n \leftarrow 0$
7 $d \leftarrow 1$
8 $S \leftarrow []$

9 **for** $i \leftarrow 0$ **to** $n'$ **do**
10     $x \leftarrow d - n$
11     **if** $x = 1$ **then**
12         $n \leftarrow 1$
13         $d \leftarrow d \times b$
14     **end**
15     **else**
16         $y \leftarrow d \div b$
17         **while** $y \geq x$ **do**
18             $y \leftarrow y \div b$
19         **end**
20         $n \leftarrow ((b + 1) \times y) - x$
21     **end**
22     $S$.append($n \div d$)
23 **end**

24 **return** $S$

---

8. Photo of a young woman with long, wavy brown hair tied in a bun and glasses. She has a fair complexion and is wearing subtle makeup, emphasizing her eyes and lips. She is dressed in a black top. The background appears to be an urban setting with a building facade, and the sunlight casts a warm glow on her face.

9. Photo of a young man in a black suit, white shirt, and black tie. He has a neatly styled haircut and is looking directly at the camera with a neutral expression. The background consists of a textured wall with horizontal lines. The photograph is in black and white, emphasizing contrasts and shadows. The man appears to be in his late twenties or early thirties, with fair skin and short, dark hair.

10. Selfie photo of a wizard with a long beard and purple robes, he is apparently in the middle of Tokyo. Probably taken from a phone.

11. An image of Pikachu enjoying an elegant five-star meal with a breathtaking view of the Eiffel Tower during a golden sunset.

12. A sleek airplane soaring above the clouds during a vibrant sunset, with a stunning view of the horizon.

13. A towering mecha robot overlooking a vibrant favela, painted in bold, abstract expressionist style.

14. Anime art of a steampunk inventor in their workshop, surrounded by gears, gadgets, and steam. He is holding a blue potion and a red potion, one in each hand

15. Pirate ship trapped in a cosmic maelstrom nebula rendered in cosmic beach whirlpool engine.

16. A futuristic solarpunk utopia integrated into the lush Amazon rainforest, glowing with advanced technology and harmonious nature.

17. A teddy bear wearing a blue ribbon taking a selfie in a small boat in the center of a lake.

18. Digital art, portrait of an anthropomorphic roaring Tiger warrior with full armor, close up in the middle of a battle.

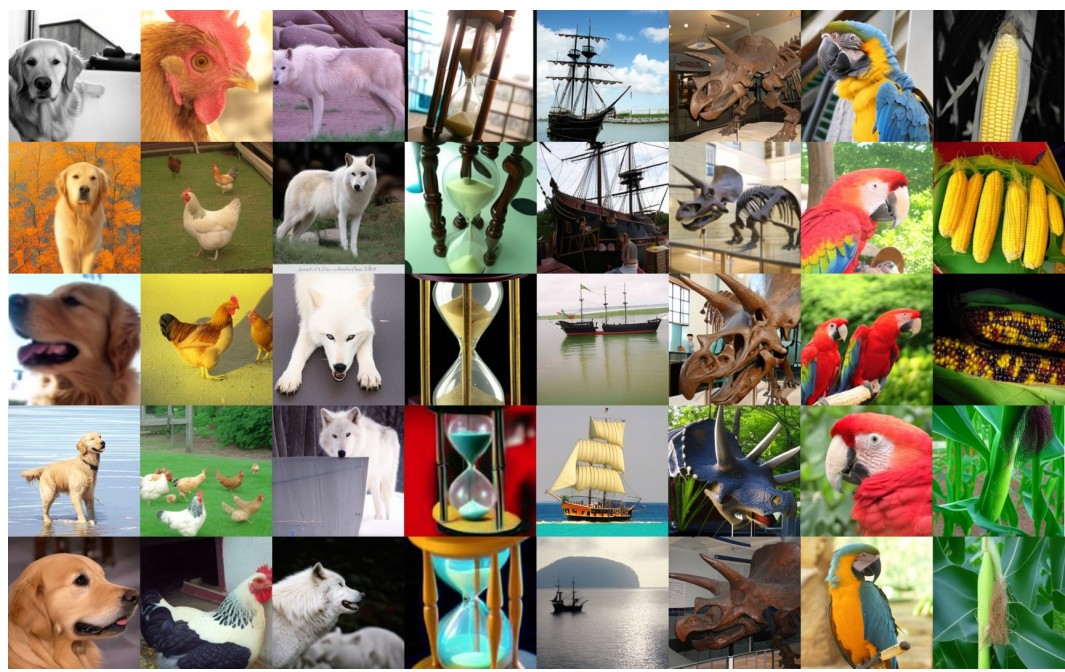

(a) **MaskGIT using our Halton scheduler.**

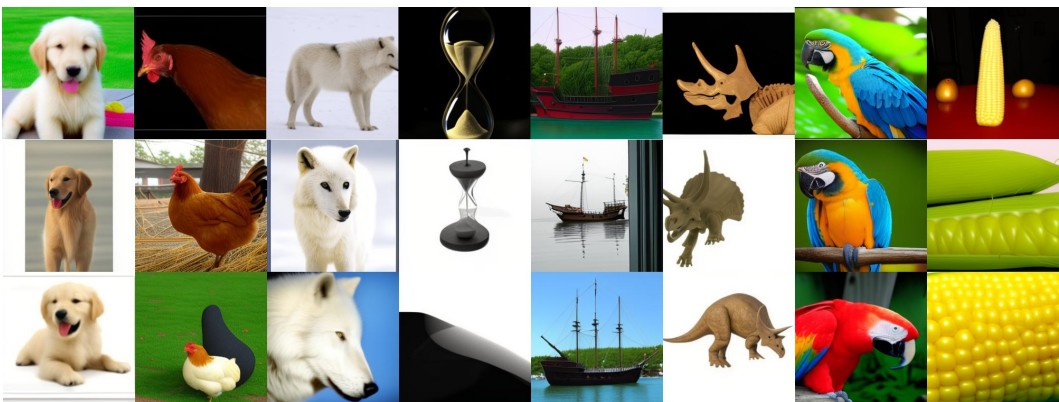

(b) **MaskGIT using the Confidence scheduler.**

Figure 11: **Scheduler comparison on random samples generated by a class-to-image model.** The Halton scheduler demonstrates a higher level of detail, capturing finer features than the Confidence scheduler, which lacks details, especially in the background.

## 12 Random Samples from our class conditioned model

In Figure 11, we show that our model can generate diverse images and more intricate details compared to the confidence scheduler. Furthermore, a comparison with the Confidence sampler reveals that the latter produces overly simplistic and smooth images, often with poorly defined backgrounds. In contrast, our approach consistently produces greater diversity, particularly in rendering background elements.

## 13 Failure Cases

**Multiple Objects.** The initial tokens sampled by the Halton scheduler tend to spread across the image, leading to instances where multiple objects or entities appear within a single image. For

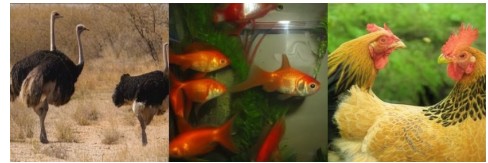

(a) **Multiple Object Generation**

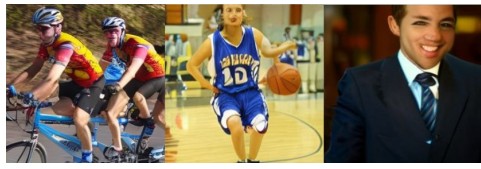

(b) **Human attributes**

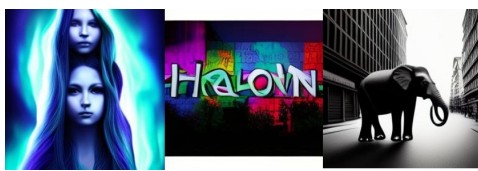

(c) **Prompt adherence:**
*- A female character with long, flowing hair that appears to be made of ethereal, swirling patterns resembling the NL...*
*- A vibrant street wall covered in colorful graffiti, the centerpiece spells 'HALTON'.*
*- An elephant is riding a bicycle in an empty street.*

Figure 12: **Failure cases.** The Halton scheduler solves some, but not all the challenges of sampling tokens in parallel. Long-range correlations still pose a challenge for MaskGIT with the Halton Scheduler.

example, this can result in multiple occurrences of a specific object, such as multiple goldfish, or even multiple parts of the same entity, such as a bird with two heads, see Figure 12a for class conditioning and Figure 12c for text-to-image.

**Inability to Self-Correct.** Unlike diffusion models, MIM-based methods cannot iteratively correct earlier predictions. Diffusion models generate predictions over the entire image at each step, allowing for refinement and correction of previous errors. In contrast, once a token is predicted in MIM, it remains fixed, even if incorrect, as there is no mechanism for subsequent correction during the generation process.

**Challenges in Complex Class/Prompt.** The model exhibits difficulties in generating certain complex classes and adhering closely to prompts. As demonstrated in Figure 12b, the model struggles to accurately generate human faces or bodies in ImageNet.

Similarly, in text-to-image conditioned tasks, it can fail to produce coherent scene compositions or faithfully render text, especially when dealing with intricate or abstract descriptions. Indeed, the model fails to render the word "HALTON" in Figure 12c and the bicycle below the elephant for the last image.

