# OpenReview forum: "Halton Scheduler for Masked Generative Image Transformer"
_ICLR.cc/2025/Conference — ICLR 2025 Poster_

### Official Review · Reviewer_13LS · 2024-10-30

**Soundness:** 1
**Presentation:** 1
**Contribution:** 1
**Rating:** 6
**Confidence:** 5

**Summary:**

This paper proposed a new sampling strategy using the Halton scheduler which uniformly recovers the image spatially. Several experiments are conducted to validate the effectiveness.

**Additional Comments**:

The manuscript does not follow the margin of ICLR template, which violates the submission guidelines.

In all, I lean towards desk rej.

**Strengths:**

NA

**Weaknesses:**

- The paper is poorly written. There are tons of typos and layout issues making it hard to follow.
- The contribution is not enough. It is like a simple spatial prior to the sampling process.
- The experiments are limited and unfair. For example, in table 1, the author compared the proposed method with the baseline MaskGIT. However, the "Ours" baseline is not presented. The compared MaskGIT has a different # param compared to the proposed approach.
- Fig 5 is very confusing. It is not understandable. Why is only the FID/IS reported for AR mode?
- In table 2, the results are not fair. As compared MaskGIT* is not trained to predict images in 32 steps.
- The qualitative results are also limited. What are the text prompts for fig 8?
- Fig 7 is very confusing. Is the training target of the confidence scheduler set to be 32 during training?

**Questions:**

NA

---

> ### Comment · Reviewer_13LS · 2024-11-22
>
> I thank the authors for their reply. However, I don't think I have a "clear misunderstanding". To clarify, I carefully read your paper and I am very familiar with MIM-based image generation.
>
> - Severe margin modification means not following the template. In addition, ICLR requires the appendix to be after the bibliography. The author should read and follow the submission guidelines.
>
> - In addition, as described in the weakness, the experiments are completely unfair. In the rebuttal, the authors mentioned Table 2. I would like to recover the hidden information that is unfair in Table 2 and bold them here.
>
> | Models                          | Scheduler   | Steps ↓ | FID ↓ | IS ↑   | Precision ↑ | Recall ↑ | Model Size | Resolution |
> |---------------------------------|-------------|---------|-------|--------|-------------|----------|-------------|----------|
> | MaskGIT (official)   | Confidence  | **12**      | 7.32  | 156.0  | 0.78        | 0.50     | 227M | **512** |
> | MaskGIT* (reproduced)                                | Confidence  | 32      | 8.38  | 180.6  | 0.79        | 0.49     | 246M | 256 |
> | | Random      | 32      | 10.47 | 174.3  | 0.75        | 0.52     | 246M | 256 |
> |                                 | Halton (ours)| 32     | 6.11 | 184.0 | 0.80   | 0.57 | 246M | 256 |
> | **Ours‡**                       | Confidence  | 32      | 5.93  | 301.1 | 0.87   | 0.46     | **705M** | 256 |
> |                                 | Random      | 32      | 6.33  | 276.0  | 0.73        | 0.49     | **705M** | 256 |
> |                                 | Halton (ours)| 32     | 3.74 | 279.5 | 0.81        | 0.60 | **705M** | 256 |
>
> The author follows a reproduced 32-step scheduling during sampling. However, the original MaskGIT leverages 8 or 12 steps to sample and achieved much better results compared to the reproduced version. To validate the effectiveness, the authors should follow the original implementation of MaskGIT (which is publicly available) or use a fair setting to demonstrate the baseline is carefully constructed.
>
> For now, it is not clear whether or not the improvement can still be achieved with a well-trained MaskGIT.
>
> ---
>
> In all, even without considering the formatting issues, the paper should also be rejected. I suggest that the authors carefully reorganize the experiments under fair conditions and explicitly disclose all settings in the manuscript for revision."

---

> > ### Author Response · Authors · 2024-11-23
> >
> > We respectfully draw the reviewer's attention on Table 2, as the resolution for the models you have reported is incorrect and the purpose of this table appears to have been misunderstood. The primary intent of Table 2 is not to compare models-vs-models directly—this comparison is the focus of Table 1. Instead, Table 2 aims to illustrate the impact of different configurations including different numbers of parameters for model, steps and implementation, on the **scheduler**. Specifically, we provide two additional rows:
> >    - (1) a version of MaskGIT* (reproduced) with a smaller number of steps (12)
> >    - (2) a version of our own MaskGIT with a reduced number of parameters
> >
> > | **Models**                | **Scheduler** | **Steps ↓** | **FID ↓** | **IS ↑** | **Precision ↑** | **Recall ↑** | **Model Size** | **Resolution**   |
> > | ------------------------- | --------------- | ------------- | ----------- | ---------- | ----------------- | -------------- | ---------------- | ------------------ |
> > | MaskGIT (official)        | Confidence    | 12          | 7.32      | 156.0    | 0.78            | 0.50         | 227M           | 512              |
> > |---------------------------|---------------|-------------|-----------|----------|-----------------|--------------|----------------| ---------------- |
> > | MaskGIT* (reproduced)     | Confidence    | 12          | 7.6       | 185.1    | 0.81            | 0.49         | 246M           | 512              |
> > |                           | Random        | 12          | 18.84     | 132.1    | 0.69            | 0.48         | 246M           | 512              |
> > |                           | Halton (ours) | 12          | 7.1       | 186.4    | 0.80            | 0.55         | 246M           | 512              |
> > |---------------------------|---------------|-------------|-----------|----------|-----------------|--------------|----------------| ---------------- |
> > | MaskGIT* (reproduced)     | Confidence    | 32          | 8.38      | 180.6    | 0.79            | 0.49         | 246M           | 512              |
> > |                           | Random        | 32          | 10.47     | 174.3    | 0.75            | 0.52         | 246M           | 512              |
> > |                           | Halton (ours) | 32          | 6.11      | 184.0    | 0.80            | 0.57         | 246M           | 512              |
> > |---------------------------|---------------|-------------|-----------|----------|-----------------|--------------|----------------| ---------------- |
> > | Ours‡                     | Confidence    | 32          | 7.5       | 227.2    | 0.76            | 0.56         | 484M           | 256              |
> > |                           | Random        | 32          | 9.7       | 196.2    | 0.75            | 0.51         | 484M           | 256              |
> > |                           | Halton (our)  | 32          | 5.3       | 229.2    | 0.76            | 0.63         | 484M           | 256              |
> > |---------------------------|---------------|-------------|-----------|----------|-----------------|--------------|----------------| ---------------- |
> > | Ours‡                     | Confidence    | 32          | 5.93      | 301.1    | 0.87            | 0.46         | 705M           | 256              |
> > |                           | Random        | 32          | 6.33      | 276.0    | 0.73            | 0.49         | 705M           | 256              |
> > |                           | Halton (our)  | 32          | 3.74      | 279.5    | 0.81            | 0.60         | 705M           | 256              |
> > |---------------------------|---------------|-------------|-----------|----------|-----------------|--------------|----------------| ---------------- |
> >
> > Furthermore, for additional insights into the model's behavior based on the number of steps, we have already presented this analysis in Figure 7 of the paper.

---

> > > ### Comment · Reviewer_13LS · 2024-11-25
> > >
> > > I am increasing my rating to 6 for the revised version.
> > >
> > > The initial submission had major issues but the authors managed to address most of them during the rebuttal. The experiments can still be improved by enrolling settings that are more comparable to previous baselines. There exists a potential risk for accepting the manuscript. As shown in the updated Table 2, with a more fair 12-step inference schedule, the performance gain dropped from +2.27 to +0.45 FID. When compared with the official 12-step setting (7.32 FID, smaller -19M model size), the gain is further shrunk to +0.17 FID.
> > >
> > > One last suggestion, for the system-level comparison in Table 1, it would be better to replace "Ours+halton" with other words, such as "MaskGIT*+Ours" to describe MaskGIT modified by the authors (though the caption clarified this). MaskGIT is not the proposed approach by the authors and thus should not be described by ours.

---

### Official Review · Reviewer_ozt1 · 2024-10-31

**Soundness:** 3
**Presentation:** 3
**Contribution:** 2
**Rating:** 6
**Confidence:** 4

**Summary:**

This paper proposes a novel scheduler based on the Halton sequence, aimed at minimizing the correlation between simultaneously sampled tokens and maximizing the information gained at each step. By selecting token positions using a quasi-random, low-discrepancy Halton sequence, the approach reduces FID scores and improves both image quality and diversity.

**Strengths:**

- Presents an innovative mutual information analysis that clarifies the scheduler’s role in MaskGIT.
- By leveraging the Halton sequence, the method ensures a more even distribution of selected tokens, grounded in the information gained at each step and the mutual information between tokens.
- Enhances both image quality and diversity in class-to-image and text-to-image generation.

**Weaknesses:**

- The assumption for Eq. 5.b is not sufficient, which ignores the long-distance dependencies.
- The influence of conditional entropy is not limited to spatial proximity; semantic information and context play a significant role and should be considered.

**Questions:**

- After vector quantization, do token relationships still correspond to their Euclidean distance?
- Might this uniformity lead to the overlooking of critical details in specific areas, thus negatively impacting sampling accuracy?

---

### Official Review · Reviewer_ogcs · 2024-11-02

**Soundness:** 3
**Presentation:** 2
**Contribution:** 3
**Rating:** 6
**Confidence:** 4

**Summary:**

This paper presents the Halton Scheduler, which refines the token unmasking strategy in MaskGIT. The authors begin by analyzing the Confidence Scheduler previously designed for MaskGIT (Chang et al., 2022; 2023), which prioritizes the unmasking of the most certain tokens. They demonstrate that while this method enhances confidence, it can reduce diversity and image quality due to the clustering of tokens around already predicted regions. To address this, they propose a novel, deterministic scheduler that leverages the Halton low-discrepancy sequence (Halton, 1964). This approach is informed by understanding entropy progression during image generation, aiming for a more even distribution of token selection to ensure uniform coverage across the image. They show convincing results over baselines like MUSE and MaskGIT across image generation and text to image generation.

**Strengths:**

1) Overall the idea of Halton scheduler makes a lot of sense and addresses the important problem of scheduler in MaskGIT kind of models.
2) Halton scheduler proposes a improved schedule which makes the model improved generation capabilities.
3) Experimental results show pretty convincing improvements across text-to-image zero-shot COCO, MUSE , MaskGIT and other competetive methods.

**Weaknesses:**

Weakness:
1) How was the discretization of the sequence Hnh into a grid corresponding to the token map done? This seems to be an approximation which I’m not sure about.
2) More intuition and details need to be added especially for Halton scheduler, like the Low-Discrepancy properties need to be discussed more.
3) Discussion on improved halton sequencing needs to be added. Using the Scrambled Halton Sequence, the Leapfrog Method, and Higher-Dimensional Generalizations need to be discussed.
4) Similar ideas for information overload into tokens have been studied in the registers[1]. Analysis on token norm in this setting will be useful and could reflect whether there is similarity in understanding between high-norm tokens and the proposed scheduler.
5) Is Extra compute time required to use Halton scheduler? One of the main advantages of the Halton scheduler is the speed of image generation. Would like to see what is the inference time with Halton's scheduler?
References
1) Vision Transformer needs registers.

**Questions:**

I don't have a good sense of what is the extra information that is being learnt using Halton schduler and the intuition is not very clear. I agree with the analysis of the paper but the proposed solution doesn't;t seem very well analysed and ablated.

---

### Official Review · Reviewer_W6JM · 2024-11-03

**Soundness:** 3
**Presentation:** 3
**Contribution:** 3
**Rating:** 6
**Confidence:** 3

**Summary:**

This paper proposes a new schedule for token decoding in masked image modeling. MaskGIT decodes high confidence tokens first. Instead, this paper shows that this technique is not optimal and can lead to decreased performance with more decoding steps which should not occur with a good schedule. To combat this, the paper proposes a Halton schedule based on mutual information analysis to spread out the predicted tokens. The technique requires no training and can be plugged into other models. Experiments are conducted on class conditional and text conditional image generation.

**Strengths:**

This paper addresses an overlooked problem: choosing which tokens to decode during masked image modeling. The intermediate analysis and motivation are clear, and the proposed solution is promising while being simple. Comparison against random decoding is also conducted.

**Weaknesses:**

I generally like this approach. The problem and motivation are clear, and the solution is simple and seems promising. However, some of the results are not convincing enough, and the presentation of the paper is unpolished and could benefit from a few rounds of editing.

- It is unclear to me how to interpret Table 1, because the baseline is from the paper for which no code exists. The models are very different parameter sizes, comparison against random and confidence is missing. I would like to see more rows like in Table 2. In fact, Table 2 and 3 are more convincingly demonstrate the effectiveness of this approach. However, the image resolutions do not match and so it is really difficult to draw conclusions.
- In general, the experimental part of this paper is weak, because if true, much more models and tasks should have been evaluated to showcase the plug&play-ability of this approach for various MIM tasks and models. From the current results, it is not clear how well the approach generalizes across datasets, models, and tasks.
- Does the method have any limitations, weaknesses or failure cases? What about runtime and memory complexity compared to random and confidence scheduling?
- Fig 5 should not be a line plot in my opinion as the x-axis is categorical.

In summary, the method is promising, and the paper presents an initial good signal. I encourage the authors to work on a more comprehensive evaluation of MIM models, tasks, and datasets to demonstrate the benefits of this model and produce fair and clear comparisons on fixed settings. As of now, unfortunately, there is not enough evidence to support the claims made convincingly.

**Questions:**

NA

---

### Meta-Review · Area_Chair_r2ZX · 2024-12-16

**Metareview:**

This paper introduces Halton scheduler as a new scheduler for Masked Generative Image Transformers' token decoding, and validates proposed approach on class-conditioned and text-conditioned image generation.

The manuscript was reviewed by four knowledgeable referees. The reviewers acknowledged that the problem tackled in the paper is well motivated (W6JM) and important (ogcs), and the solution is sound, simple and promising (W6JM, ogcs).


The main concerns raised by the reviewers were:
1. The presentation of the paper which could be significantly improved (W6JM, ogcs,13LS)
2. The experiments did not appear convincing (W6JM, ogcs, 13LS) -- e.g. comparisons made with different model sizes and image resolutions, limited number of models considered
3. The contribution seemed incremental (13LS)
4. Assumption of Eq 5 did not appear sufficient, and as a result, it was difficult to understand whether the significance of the improvements could be fully attributed to the proposed approach (ozt1)
5. The limitations of the method were not discussed (W6JM)

The authors addressed most of the above mentioned concerns during rebuttal and discussion period. The authors discussed limitations, updated the paper presentation, argued for the novelty of the proposed approach, validated the assumptions in practice, clarified their experimental setup and conclusions draw from the results. The authors addressed most questions including those related to computation time. During discussion, the authors shared with the AC their concerns w.r.t. some of the comments made in the reviews. After that, both authors and reviewer engaged in fruitful discussion.

During reviewers' discussions, reviewers agree that most of their concerns were addressed by the authors and that the paper could be interesting for the research community. Although some reviewers argue that experiments could still be improved, they lean towards acceptance. The MR agrees with the reviewers' assessment and recommends to accept.

**Additional Comments On Reviewer Discussion:**

See meta-review.

---

### Decision · Program_Chairs · 2025-01-22

Accept (Poster)